# Kondo signatures in defected Dirac spin liquids:
# Non-Abelian bosonization after Chern-Simons fermionization

Huanhuan Jiang,[1] Feilong Luo,[1] Yilin Wang,[2] Peng Song,[3, 1, 4, *] and Rui Wang[1, 4, 5, †]

[1] National Laboratory of Solid State Microstructures and Department of Physics, Nanjing University, Nanjing 210093, China
[2] Hefei National Research Center for Interdisciplinary Sciences at the Microscale,
University of Science and Technology of China, Hefei 230026, China
[3] Joint School of National University of Singapore and Tianjin University,
International Campus of Tianjin University, Binhai New City, Fuzhou 350207, China
[4] Collaborative Innovation Center for Advanced Microstructures, Nanjing 210093, China
[5] Hefei National Laboratory, Hefei 230088, China

The quantum impurity, which serves as in-situ probe of the thermal baths, is a prominent topic in materials research. Its application to quantum spin liquids has attracted great interest in last decades. However, the comprehensive understanding of the quantum impurity effects in quantum spin liquids is still an open question, due to the complicated gauge fluctuations and the strong-correlation between the fractionalized excitations and the impurity. Here, we propose a general method, a combination of the Chern-Simons fermionization and the Wess-Zumino-Witten theory. Our method shows that quantum spin liquids with local defects can induce emergent impurity phenomena, and provides a systematic solution to the quantum impurity problem in an important class of quantum spin liquids, i.e., the Dirac spin liquids. Under the Chern-Simons fermionization, the gauge fluctuations are apparently suppressed, and the strong-correlation between the fractionalized excitations and the impurity can be exactly solved by the non-Abelian bosonization. Consequently, the Fermi liquid and non-Fermi liquid fixed point as well as the crossover between them are identified, respectively, depending on the relevance of the impurity scattering among the Dirac valleys. The obtained fixed points lead to several new experimental fingerprints for Dirac spin liquids, including a Kondo-induced magneto-thermal effect, a non-monotonous thermal conductivity during the crossover, and an anisotropic spin correlation function. These findings provide a theoretical framework as well as the experimental guidance to explore novel Kondo phenomena in quantum spin liquids.

*Introduction.–* Quantum spin liquids (QSLs) [1–3], strongly entangled quantum states that evade ordering down to zero temperature, pose a great challenge for their experimental observation. The main difficulty is due to the fact that the low-lying excitations of QSLs are fractionalized particles [2, 3], whose nonlocal nature is beyond the capabilities of usual experimental probes. For a number of candidate materials, the absence of ordering is evidenced by the specific heat and the muon spin relaxation experiments at ultra-low-temperature [4–28]. However, the identification of the highly-entangled liquid states remains elusive [28–32]: the observed thermal conductivity at low temperatures suggests a dominant role played by the phonons, obscuring the contributions from the fractionalized particles [31, 32], e.g., the spinons, if any. Therefore, to further validate the QSL ground states, it is urgent to predict more finger-print experimental features that are unique to the fractionalized excitations [2].

An important strategy is to use quantum impurities as in-situ probes [33–39], which can induce many-body Kondo resonance and result in global change of the thermal dynamical properties of the bath. The most common impurities come from the perturbation of external

dopants, either magnetic or non-magnetic. These can be referred to as extrinsic impurities, and have drawn growing attention and were applied to various QSLs since the last decade [40–42], including the Kitaev spin liquids [43–46], the spin liquids with spinon Fermi surfaces [47, 48], and also the deconfined quantum critical point (DQCP) in frustrated magnets [49–52]. The Kondo signature in spinon bath was also observed by recent experiments on Zn-brochantite [53, 54]. In contrast to the extrinsic impurities, intrinsic defects are much less studied in QSLs. The intrinsic impurities introduce no external degrees of freedom to the bath, and they can appear in different forms, e.g., a local lattice distortion in candidate materials of QSLs, or a local quench of the spin exchange interactions in frustrated magnets. The intrinsic defects can also exhibit novel properties as a result of the coupling to the host material, e.g., lattice dislocations in honeycomb lattices can generate effective magnetic moments [55]. Their effects in QSLs still remain elusive to date.

For both extrinsic and intrinsic impurities, there are several key questions yet to be addressed in QSLs. First, the fractionalized excitations in QSLs are usually coupled to emergent gauge fields [41, 47, 56], but it is still not clear whether the gauge fields would participate in the local many-body resonance triggered by the impurity [47, 57]. Second, the conventional parton mean-field theory for QSLs is subject to the single-occupation condition [58], and therefore inevitably requires approxima-

———
* songpeng@tjufz.org.cn
† rwang89@nju.edu.cn

tions such as the large-N treatment when an impurity is involved. Therefore, a more rigorous approach is highly desired, which hopefully can generate novel Kondo signatures for QSLs. Third, for the intrinsic impurities, such as a local quench of the spin-exchange interaction in frustrated magnets, their effect is strongly dependent on the hosting materials. Hence, it is intriguing to ask whether they could generate novel many-body resonance in a QSL.

In this work, we propose a new analytic approach - namely, the Chern-Simons fermionization plus Wess-Zumino-Witten (WZW) theory - to systematically study the quantum impurity problems in QSLs. Unlike the conventional studies on the extrinsic impurities, we focus on the intrinsic defects and reveal their salient many-body features in a QSL. For demonstration, we propose a prototypical model describing a 2D XY (frustrated) quantum magnet with a local twist of the exchange interaction, which we term the locally twisted XY model (LTXY). Using our proposed method, we show that the local twist in the frustrated XY magnets leads to a low-energy effective theory, which describes an emergent Anderson-type impurity in a Dirac QSL. Novel Kondo phenomena are then obtained, despite their different origin compared to the conventional magnetic impurity problems.

Our study shows that an intrinsic non-magnetic defect in Dirac QSL can also generate emergent Kondo phenomena. The Dirac QSLs, whose fractionalized excitations enjoy linear dispersion, are of particular importance as they are closely related to the quantum antiferromagnetism as well as the DQCP [59]. They are also proposed as stable ground states of certain quantum spin models [60–62]. Technically, we fermionize the LTXY model using the lattice Chern-Simons (CS) fermion representation [63–67]. The obtained low-energy physics corresponds to a number of Dirac valleys with valley-dependent pseudospin-momentum locking (PSML). The local twist and its coupling to the rest of the system is then mapped to an Anderson impurity, which is coupled to the Dirac fermions in low-energy. The low-energy effective model, owing to its rotation symmetry, can be reduced to $(1 + 1)$D conformal field theories (CFTs), namely WZW theories. Then, based on the non-Abelian bosonization, two types of Kondo fixed points, either the Fermi liquid (FL) or the non-Fermi liquid (NFL), are identified, depending on the relevance of the impurity scattering among the Dirac valleys. Remarkably, we show that, although both of the two fixed points are charge-insulating, they display novel magneto-thermal conductivities with distinct scaling behaviors at low temperature. This property defines new Kondo phenomena in spin liquids generated by intrinsic defects, and allows experimental detection of Dirac QSLs with fractionalized excitations.

*A general reduction to CFT.–* Let us begin with considering the general Kondo problem with a bath of Dirac fermions: 2D Dirac valleys labeled by $a$ related by some

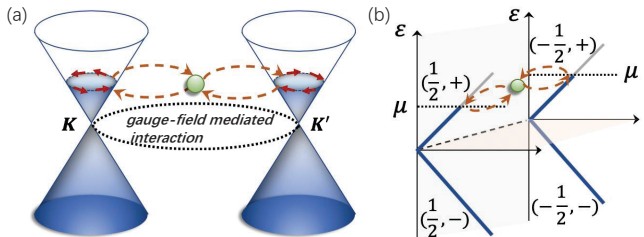

FIG. 1. (a) The impurity is coupled to different Dirac valleys, whose Fermi levels are modulated by external magnetic field (see below). (b) The low-energy modes of Dirac valleys correspond to chiral fermions with orbital angular momentum and pseudospin indices $(j, s)$. The impurity is only coupled to the soft modes with $s = +$ ($s = -$) and $j = \pm 1/2$ for a positive (negative) chemical potential.

point group [68] in the Brillouin zone, whose low-energy excitations are described by

$$H^D = \sum_a \int \frac{d^2 k}{(2\pi)^2} \, f^{a\dagger}(\boldsymbol{k})(v_F \boldsymbol{k} \cdot \boldsymbol{\tau}^{(a)} - \mu) \, f^a(\boldsymbol{k}). \quad (1)$$

Here, we allow a valley-dependent PSML, such that for each valley $a$ ($a = 1, 2, ..., k$), the set of pseudospin $\tau_i^{(a)}$ with $i = 1, 2$ can be different and are not necessarily the standard Pauli matrices. Since they satisfy the Clifford algebra, $\{\tau_i^{(a)}, \tau_j^{(a)}\} = 2\delta_{ij}\mathbb{1}_2$, we can always appropriately choose $\tau_3^{(a)}$, so that there is a unitary transformation $U^a$ that transforms $\tau_i^{(a)}$ into the standard Pauli matrices $\tau_i = U^a \tau_i^{(a)} U^{a\dagger}$. We further consider an impurity effectively characterized by $\boldsymbol{S}_{imp}$ pinned at $\boldsymbol{r} = 0$ in real space. These Dirac fermions have an effective Kondo exchange coupling with the impurity $\boldsymbol{S}_{imp}$,

$$H' = \sum_{a,b} \lambda_{ab} \, f^{a\dagger}(0)\frac{\boldsymbol{\tau}}{2} f^b(0) \cdot \boldsymbol{S}_{imp}, \quad (2)$$

where $\boldsymbol{\tau} = (\tau_1, \tau_2, \tau_3)$ and $\lambda$ is a symmetric real matrix. The point group symmetry that relates the Dirac valleys imposes constraints on $\lambda$. For instance, diagonal entries are all equal. As we observe, for the Hamiltonian in Eq.(1), the density of states (DOS) vanish at $\mu = 0$. At this critical point, the Kondo exchange coupling in Eq.(2) is irrelevant. Away from the critical point with $\mu \neq 0$, the low-energy degrees of freedom are described by the soft fermionic modes in the vicinity of the Fermi circles (see Fig.1(a)).

For technical convenience, let us transform the pseudospin $\boldsymbol{\tau}^{(a)}$ for each valley $a$ into the standard Pauli matrices. That is, we introduce $\tilde{f}^a(\boldsymbol{k}) = U^{a\dagger} f^a(\boldsymbol{k})$. In accord with the Kondo exchange Eq.(2), it is convenient to work with the polar coordinates. Since the PSML of (1) preserves the total angular momentum $J_z = L_z + \frac{1}{2}\tau_z$, we expand the fields as, $\tilde{f}^a(\boldsymbol{k}) = \sum_{j,s} f_{j,s}^a(k)\chi_{j,s}(\phi)$, where $\chi_{j,s}(\phi)$ is the eigenstates of $J_z$, with quantum numbers $(j, s)$, where $j$ is the half-integer eigenvalues of $J_z$ and

$s = \pm$ labels the eigenvalues of $\tau_z$. In such basis, incoming and out-going radial waves emerge with positive and negative energies, respectively (see Fig.1(b)). Then, for $\mu > 0$, the soft modes are right-handed fermions described by

$$H_{eff}^D = v_F \sum_{a,j} \int_{-\Lambda}^{\Lambda} \frac{dq}{2\pi} \; \psi_j^{a\dagger}(q) q \psi_j^a(q), \qquad (3)$$

where $\psi_j^a(q) = u_j^a \sqrt{\frac{k_F}{2\pi}} \tilde{f}_{j,+}(k_F + q)$ with the index $s = +$ omitted. Here, an arbitrary $U(1)$ factor freedom $u_j^a$ is allowed for each soft mode $(a,j)$ for later usage. The Fermi momentum is $k_F = \mu/v_F$, and $q$ takes value within a cutoff, $q \in [-\Lambda, \Lambda]$. For $\mu < 0$, analogously the soft modes corresponds to left-handed fermions, and can be treated in parallel to the case of $\mu > 0$. Note that the renormalized soft fermion fields by $\sqrt{\frac{k_F}{2\pi}}$ satisfy the anti-commutation relations of 1D fermions, for instance, $\{\psi_j^\dagger(q), \psi_{j'}(q')\} = 2\pi\delta(q - q')$.

The Kondo exchange $H'$ can be transformed accordingly, reducing to the coupling to 1D soft fermions. Owing to the $U(1)$ factor $u_j^a$ of the soft modes, it can be cast into a simple form where only $j = \pm 1/2$ are relevant:

$$H_{eff}' = \sum_{a,b} g_{ab} \; \psi^{a\dagger}(0) \frac{\boldsymbol{\sigma}}{2} \psi^b(0) \cdot \boldsymbol{S}_{imp}, \qquad (4)$$

where $\psi^{a\dagger}(x) = [\psi_{\frac{1}{2}}^{a\dagger}(x), \psi_{-\frac{1}{2}}^{a\dagger}(x)]$, $g_{ab} = \pi k_F \lambda_{ab}$, and $\boldsymbol{\sigma}$ denotes the Pauli matrix defined in the angular momentum space. Eq.(4) implies a cutoff, with summation over only $j = \pm 1/2$ in Eq.(3).

The above derivation shows that general Kondo problems in 2D Dirac systems with valley-dependent PSML can be reduced to $k$ valleys of soft fermions coupled to the impurity, which generally allows for a CFT description of the underlying infared fixed points [69–76]. The single valley case is illustrated by Fig.1(b).

Let us firstly consider the case with ignorable intervalley scatterings, namely $g_{ab} = g\delta_{ab}$. Then, the Kondo exchange Eq. (4) becomes $H_{eff}' = g\boldsymbol{J}(0) \cdot \boldsymbol{S}_{imp}$, where the $SU(2)$ current $\boldsymbol{J}(x) = \sum_a \psi_a^\dagger(x)\boldsymbol{\sigma}\psi_a(x)/2$. This motivates us to consider the global symmetry $U(1) \times SU(k) \times SU(2)$ of Eq.(3) for the charge, valley and pseudospin sector, separately, leading to the bosonization for the Hamiltonian density as [76],

$$\mathcal{H}_{eff}^D = \frac{\pi v_F}{2k} J^2 + \frac{2\pi v_F}{k+2} \boldsymbol{J}^2 + \frac{2\pi v_F}{k+2} \mathcal{J}^2, \qquad (5)$$

where the currents for $U(1)$ and $SU(k)$ sector respectively read as, $J = \sum_{a,j} \psi_j^{a\dagger}\psi_j^a$ and $\mathcal{J}^A = \sum_j \psi_j^\dagger T^A \psi_j$, with $T^A$ the generators of $SU(k)$. Since the impurity only interacts with the pseudospin current operator $\boldsymbol{J}$, which satisfies the Kac-Moody algebras $SU(2)_k$, the fermion bath can enjoy a NFL fixed point [69, 70] characterized by $U(1) \times SU(2)_k \times SU(k)_2$ CFT. In the generic case with off-diagonal entries of $g$, the Kondo exchange Eq.(4) violates

the rotational symmetry in the valley space. Therefore, in general $\mathcal{J}$ is no longer a conserved current, and the level of $SU(2)_k$ will be reduced, resulting in a different fixed point.

*Chern-Simons Dirac fermions in spin liquids.*– We now demonstrate how the above formalism can be related to a general 2D Dirac QSL. Our scheme is to utilize the CS fermionization [63–67] to describe the Dirac QSLs. We represent the local spin-1/2 state as a spinless fermion state attached with a unit of $U(1)$ gauge flux to preserve the bosonic statistics, or equivalently in terms of operators, $S_{\mathbf{r}}^\pm = f_{\mathbf{r}}^\pm e^{\pm iU_{\mathbf{r}}}$ with $U_{\mathbf{r}} = \sum_{\mathbf{r}' \neq \mathbf{r}} \arg(\mathbf{r} - \mathbf{r}') f_{\mathbf{r}}^\dagger f_{\mathbf{r}}$. The flux attachment for each fermion is enabled by coupling the fermions to a $U(1)$ gauge field described by a CS term [64, 66]. Under the fermionization, the low-energy physics of a frustrated spin system can be derived as the emergent Dirac CS fermions with competing nonlocal interactions induced by gauge field [66, 67]. A gapped spin liquid is then formed when certain bosonic orders are generated [66], while the gapless Dirac QSL naturally emerges when the interaction becomes irrelevant [67].

We specify our study using the Hamilltonian for a 2D XY quantum magnet as starting point,

$$H_0 = \sum_{\mathbf{r},\mathbf{r}'} J_{\mathbf{r},\mathbf{r}'}(S_{\mathbf{r}}^x S_{\mathbf{r}'}^x + S_{\mathbf{r}}^y S_{\mathbf{r}'}^y). \qquad (6)$$

Here, we consider a honeycomb lattice as an example. For the $J_1 - J_2$ honeycomb XY model, signatures of a gapless Dirac spin liquid state are numerically revealed at the critical point $J_2/J_1 \sim 0.23$ [77]. Here, since our focus is to solve the impurity problem, the Dirac QSL ground state is assumed. $J_{\mathbf{r},\mathbf{r}'}$ includes the first several nearest neighbor (NN) interactions with frustration. Then we consider an intrinsic defect on top of Eq.(6), i.e., we replace the nearest neighbor XY exchange interaction at site $\mathbf{r}_0$ by the ZZ interaction,

$$H_{def} = J' S_{\mathbf{r}_0,A}^z S_{\mathbf{r}_0,B}^z, \qquad (7)$$

where $A$ and $B$ denote the sublattice sites $A$ and $B$ of $\mathbf{r}_0$, and $J'$ denotes the coupling coefficient between them. On the lattice bond located at $\mathbf{r}_0$, the original $S_{\mathbf{r}_0,A}^x S_{\mathbf{r}_0,B}^x + S_{\mathbf{r}_0,A}^y S_{\mathbf{r}_0,B}^y$ term has been "twisted" into $S_{\mathbf{r}_0,A}^z S_{\mathbf{r}_0,B}^z$. We term the locally defected XY model (6)+(7) the LTXY model, as schematically shown in Fig.2(a). We shall further analyze the LTXY model under CS fermionization.

After fermionization of Eq.(6), the CS fermions are cast into the same Hamiltonian as Eq.(1), with additional gauge field-mediated interactions (Fig.1(a)) [64–67]. We now focus on honeycomb lattice, where two Dirac valleys $a = \pm$ emerge at $\mathbf{K}$ and $\mathbf{K}'$, related by mirror symmetry, and accordingly $\boldsymbol{\tau}^{(+)} = \boldsymbol{\tau}$ and $\boldsymbol{\tau}^{(-)} = -\boldsymbol{\tau}^T$ are Pauli matrices defined in the pseudospin (sublattice) space [64–66], indicating the valley-dependent PSML. For other lattices, there can be more Dirac valleys related by point groups [66, 67]. We restrict ourselves to studying a stable Dirac QSL [78, 79] such that the gauge field-induced

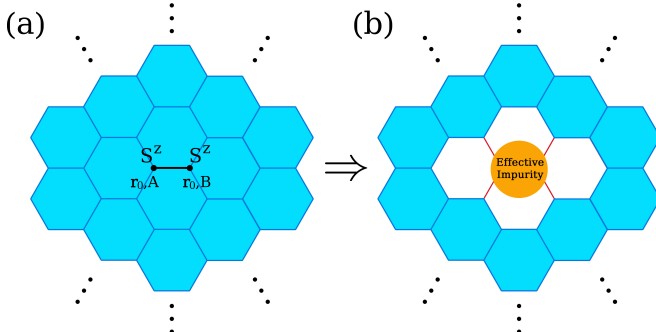

FIG. 2. (a) The LTXY model on a honeycomb lattice, which is a XY model with one of its lattice bounds $\mathbf{r}_0$ twisted into $S^z_{\mathbf{r}_0,A}S^z_{\mathbf{r}_0,B}$. (b) The LTXY model can be mapped into an effective Anderson impurity model. The local defect/twist is mapped into an effective Anderson impurity, coupled to the bath consisting of the surrounding honeycomb lattice via the nearest neighbour couplings shown as red lines.

interactions between the Dirac fermions are irrelevant operators [64].

The CS fermion representation reveals that, the chemical potential $\mu$ of the CS Dirac fermions is tunable by an out-of-plane field $B$. This is because, as long as the Dirac QSL remains stable, the field $B$ generates the out-of-plane polarization that modulates the density of CS fermions $n$ via $\sum_{\mathbf{r}}\langle S^z_{\mathbf{r}}\rangle/N = n - 1/2 \propto B$ [65, 67].

We shall show in the appendix that after CS fermionization of Eq.(7), $H_{def}$ takes the form of an effective Anderson impurity, which couples to the Dirac QSL via nearest neighbour couplings and perturbs its ground state (see Fig.2(b)). The Schrieffer-Wolff transformation then transforms the Anderson impurity model to an effective Kondo exchange model

$$H' = \sum_{\mathbf{r}} \lambda(\mathbf{r}) f^{\dagger}(\mathbf{r}) \frac{\boldsymbol{\tau}}{2} f(\mathbf{r}) \cdot \mathbf{S}_{imp}, \tag{8}$$

where we have set $\mathbf{r}_0 = 0$, and $\boldsymbol{\tau}$ denotes the pseudospin (sublattice). $\lambda(\mathbf{r})$ is the coupling strength where a $\mathbf{r}$-dependence is allowed for generality. $\mathbf{S}_{imp}$ is an effective local spin-1/2 impurity. Interestingly, the CS fermionization translates the original local defect into a Kondo exchange in the pseudospin (sublattice) space.

It is natural to assume that $\lambda(\mathbf{r})$ exponentially decays away from the impurity, namely, $\lambda(\mathbf{r}) = \lambda_0 e^{-|\mathbf{r}|/\xi}$, where $\xi$ is the characteristic scattering length. Then, with projection into the low-energy window, Eq.(8) takes the form of Eq. (2). Specifically, the diagonal and off diagonal entries of $\lambda$, $\lambda_d$ and $\lambda_t$, correspond to the intervalley and the intravalley scattering strength, respectively. They are explicitly given by $\lambda_d = 2\pi\lambda_0\xi^2$ and $\lambda_t = 2\pi\lambda_0\xi^2/(1 + |\mathbf{Q}|^2\xi^2)^{3/2}$ with $\mathbf{Q} = \mathbf{K} - \mathbf{K}'$ for $\Lambda \ll \xi^{-1}$. Here, $\lambda_t$ is vanishingly small compared to $\lambda_d$ for long-range scattering but is non-negligible for short-range scattering.

The above shows a systematic mapping from the quantum impurity model in frustrated magnet to the Kondo model in 2D Dirac fermions with valley-dependent PSML, i.e., Eqs. (1) and (2). Accordingly, the reduction to the low-energy soft modes follows, producing Eq. (4) with the diagonal and off-diagonal entries as, $g_{d/t} = \pi k_F \lambda_{d/t}$.

Before proceeding, we compare the Dirac fermion bath in the spin liquids with that in semimetals [80, 81] and the surface states of topological insulators [82]. While both have linear dispersion, the Dirac CS fermions in the spin liquids enjoy extraordinary features: First, the CS fermions are both charge-insulating and spinless. Second, the Kondo exchange acts in the sublattice rather than the true spin space. Third, the chemical potential is tunable by magnetic field, rather than by the electric potential.

*Kondo fixed points and thermal conductivity.*– For the present spin liquid with two Dirac valleys, the pseudospin and valley currents both satisfy SU(2)$_2$ algebra. The bosonization of the low-energy modes are given by Eq.(5). Accordingly, if $g_t$ is negligible, we expect that the impurity is over-screened, and the fermion bath corresponds to the NFL fixed point governed by U(1) × SU(2)$_2$ × SU(2)$_2$ CFT.

Otherwise, with nonvanishing exchange $g_t$, the rotational symmetry in the valley space will be broken. Thus, we introduce $\psi_{1,2} = (\psi_+ \pm \psi_-)/\sqrt{2}$ to diagonalize the Kondo exchange term Eq.(4) into $H'_{eff} = \sum_{\alpha=1,2} g_{\alpha} \mathbf{J}_{\alpha}(0) \cdot \mathbf{S}_{imp}$, where $\mathbf{J}_{\alpha} = \psi^{\dagger}_{\alpha} \boldsymbol{\sigma} \psi_{\alpha}/2$ with $\alpha = 1, 2$ and $g_{1,2} = g_d \pm d_t$. Accordingly, the two flavors of fermions in Eq.(1) should be bosonized individually, which leads to

$$\mathcal{H}_0^{(\alpha)} = \frac{\pi v_F}{2} J_{\alpha}^2 + \frac{2\pi v_F}{3} \mathbf{J}_{\alpha}^2. \tag{9}$$

The bosonized Hamiltonian $\mathcal{H}_0^{(\alpha)}$ suggests a FL fixed point corresponds to U(1) × SU(2)$_1$ CFT.

The above expectations from CFT can be verified by the perturbative RG calculations. To third order expansion of $g_1$ and $g_2$ (see appendix), we obtain the following RG flow, $dg_1/dl = g_1^2 - g_1(g_1^2 + g_2^2)g_1/2$ and $dg_2/dl = g_2^2 - g_2(g_1^2 + g_2^2)g_1/2$. The flow trajectory is shown in Fig.3(a), where two fixed points are revealed as indicated by the red and green dot, respectively. The green dot has one of the couplings been renormalized to zero, thereby describing a FL fixed point, while the red preserves the symmetric two-channel couplings, suggesting the NFL behavior.

Using the CFT techniques [69–71], the Green's function (GF) at Kondo fixed points can be calculated by fusion with $\mathbf{S}_{imp} = 1/2$ conformal tower. It is obtained that the quasi-particle weight of CS fermions is fully preserved and lost for the FL and NFL fixed point, respectively. The latter predicts an interesting phenomena that the fractionalized excitations in spin liquid lose their quasi-particle nature due to the competing screening channels. Furthermore, via a double fusion procedure [69–71], the scaling behavior of impurity dynamical susceptibility is obtained as $\chi(\omega) \propto \omega^0$ for NFL, while $\chi(\omega) \propto \omega$ for the FL fixed point.

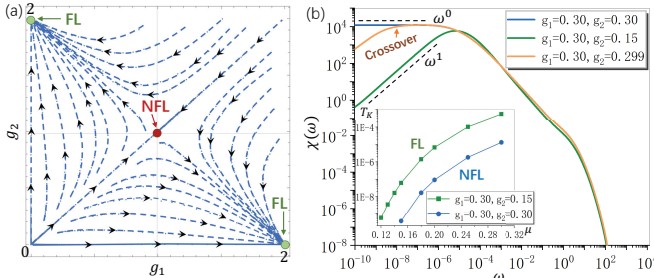

FIG. 3. (a) RG flow diagram of the coupling constants $g_1$, $g_2$, which reveal the NFL and FL fixed points. (b) The imaginary part of the impurity dynamical susceptibility of spin calculated by NRG. The NFL and FL behaviors as well as a crossover are shown in the low temperature regime for different coupling constants. The inset shows the dependence of Kondo temperatures versus chemical potential $\mu$.

To confirm the CFT results, we set up numerical renormalization group (NRG) calculations starting from the initial Hamiltonian, Eq.(1),(2). The calculations are performed using the full-density-matrix NRG [83] method implemented in the QSpace tensor library [84, 85]. As shown in Fig.3(b), for $g_1 \neq g_2$ [$g_1 = g_2$], the numerical results of dynamical susceptibility indeed shows $\chi(\omega) \propto \omega$ [$\chi(\omega) \propto \omega^0$] at low energies, clearly demonstrating the FL [NFL] fixed point. For $g_1 \sim g_2$, a crossover from NFL to FL is also found. Importantly, the inset of Fig.3(b) shows that the Kondo energy scale $T_K$ is dependent on the chemical potential of CS fermions, implying a tunable Kondo screening by external field $B$.

Some key features should be pointed out, in contrast with the FL and NFL fixed points in normal metals [74]. First, the inherent valleys of Dirac QSLs complicate the situations. The scattering potential among valleys matters. For short-range scattering, the inter-valley scattering is non-negligible, favoring the FL, whereas, the long-range scattering prefers the NFL fixed point. Second, the CS fermions carry no electron charges and are free from any resistivity anomalies [74]. However, for the FL [NFL] fixed point, the exact screening [overscreening] of the pseudospin takes pace. This should result in an anomalous thermal effect as it generates a many-body local resonance of the CS fermions.

To investigate the Kondo-generated thermal effect, we combine the CFT of CS fermions and the linear response theory (see appendix and [86]). The thermal conductivity can be calculated from the current-current correlation functions, $\pi(i\omega_n) = -\int_0^\beta d\tau e^{i\omega_n \tau} \langle \hat{T}_\tau \mathbf{j}_E(\tau) \cdot \mathbf{j}_E(0) \rangle$, where $\mathbf{j}_E(\tau)$ is the thermal current, and the thermal conductance $T\sigma_E(T) = -\lim_{\omega \to 0} \text{Im}\pi^R(\omega)/\omega$, where $\pi^R$ is the retarded correlation function obtained via analytic continuation. Further inserting the self-energy obtained from CFT, we obtain the thermal conductivity at low temperature as (see appendix),

$$\sigma_E(T)/T = \pi^3 \rho_0 /[9(1-S)n_{imp}], \quad (10)$$

where we have assumed a dilute distribution of impurities with density $n_{imp}$, and $S = -1$ [$S = 0$] for the FL [NFL] fixed point. Eq.(10) indicates that $\sigma_E(T)$ is sensitive to the DOS of CS fermions at Fermi energy, $\rho_0$, which is in turn proportional to field $B$, implying a field-modulated thermal conductivity, termed as the magneto-thermal effect. Moreover, since the phonon's contribution is field-independent, the predicted Kondo phenomena provides a controllable way to distinguish the intrinsic degrees of freedoms of QSLs. For finite temperature, the higher order corrections from the irrelevant operators in CFT come into play [74, 75], generating different scaling behaviors for the two different fixed points, i.e., $\sigma_E^{FL}(T)/T = \pi^3 \rho_0/18 n_{imp} - aT^2$ and $\sigma_E^{NFL}(T)/T = \pi^3 \rho_0/9 n_{imp} - bT^{1/2}$, where $a$, $b$ are universal coefficients. Therefore, in the crossover regime from NFL to FL shown in Fig.3(b), we expect a non-monotonous thermal conductivity $\sigma_E(T)/T$ versus $T$ when the Kondo resonance is formed.

*Conclusions and discussions.–* We have presented a general method, namely a combination of the CS fermionization [64–66] with the WZW theory, to explore novel quantum impurity effects in Dirac QSLs. Consequently, we show that effective Kondo phenomena can emerge in Dirac QSLs with local defects. FL and NFL behaviors as well as a crossover between them are found, leading to several remarkable predictions, including a Kondo-induced magneto-thermal effect in the charge-insulating state, a non-monotonous thermal conductivity during the crossover, and an anisotropic spin correlation function because of the PSML. The last one is similar to the pseudospin Kondo-singlet discussed in topological superconductors [39]. These predicted Kondo phenomena naturally provide protocols to probe the features of host Dirac QSLs. In this regard, our findings may be relevant for materials such as $\kappa - \text{ET}_2\text{Cu}_2(\text{CN})_3$ [4] and $\text{M[Pd(dmit)}_2]_2$ [29], which enjoys triangular lattice geometry and could be promising candidates for Dirac QSLs [60, 67]. Finally, we note a recent unusual field-dependence of the muon relaxation at low fields in Zn-brochantite [53]. It is an inspiring direction to investigate whether it is related to the mechanism discussed here. Our work reported here therefore is expected to have wide applications and major implications for further exploration of novel quantum impurity effects in QSLs.

## ACKNOWLEDGMENTS

We thank Andreas Weichselbaum (Brookhaven National Laboratory, USA) and Seung-Sup Lee (Ludwig Maximilian University of Munich, Germany) for providing us the QSpace tensor library and the NRG code for arbitrary type of bath, respectively. Y. W. was supported by the U.S. Department of Energy, Office of Science, Basic Energy Sciences as a part of the Computational Materials Science Program through the Center for Computational Design of Functional Strongly Corre-

lated Materials and Theoretical Spectroscopy. This work was supported by the Youth Program of National Natural Science Foundation of China (No. 11904225) and the National Key R&D Program of China (Grant No. 2017YFA0303200).

## Appendix A: The effect of the defect on the Dirac CS fermions

In this section, we show the effect of the local defect on the 2-dimensional XY quantum magnet model. Namely, the defect effectively acts as an Anderson impurity, which induces an effective Kondo exchange in the pseudo-spin (sublattice) space.

First, we need to separate the local sites at $\mathbf{r}_0$ from the rest of the system. We regard the sites at $\mathbf{r}_0$ as the intrinsic impurity and the rest of the system as the hosting bath. Thus, the bath here actually consists of a Dirac liquid but with two local sites at $\mathbf{r}_0$ forbidden. This can be well captured by introducing a large onsite potential $V$ at $\mathbf{r}_0$ [87], namely,

$$H_{bath} = H^D + V f_{\mathbf{r}_0}^\dagger f_{\mathbf{r}_0}, \quad (A1)$$

where $H^D$ is given by equation (1) of the main text, which is the 2-dimensional XY quantum magnet model $H_0$ given by equation (6) of the main text after CS fermionization, and $f_{r0} = [f_{r0,A}, f_{r0,B}]^T$. We emphasize that this large on-site potential term has little impact on the Kondo physics of our setup, especially for finite $\mu$, since its effect is only to bring about the local particle-hole asymmetry that in turn favors the Kondo fixed point, as has been proved in [88].

Next, consider the local defect $H_{def}$ given by equation (7) of the main text:

$$H_{def} = J' S_{\mathbf{r}_0,A}^z S_{\mathbf{r}_0,B}^z. \quad (A2)$$

Under CS fermionization with

$$S_{\mathbf{r}}^z = f_{\mathbf{r}}^\dagger f_{\mathbf{r}} - \frac{1}{2}, \quad (A3)$$

it is cast into

$$H_{def} \equiv H_{imp} = \sum_\alpha \epsilon_f f_{\mathbf{r}_0,\alpha}^\dagger f_{\mathbf{r}_0,\alpha} + J' f_{\mathbf{r}_0,A}^\dagger f_{\mathbf{r}_0,A} f_{\mathbf{r}_0,B}^\dagger f_{\mathbf{r}_0,B}, \quad (A4)$$

where $\alpha \in \{A, B\}$ denotes the sublattice index and $\epsilon_f = -J'/2$. Therefore, the local defect is transformed to a symmetric Anderson impurity, with the local effective CS fermions subjected to a Hubbard interaction.

Then, we consider the coupling $H_{hyb}$ between $H_{bath}$ and the effective impurity $H_{imp}$, in the form of hoppings between the impurity sites and their nearest neighboring sites in the bath:

$$H_{hyb} = \sum_{\mathbf{r}'} J_{\mathbf{r}_0,\mathbf{r}'} \left( f_{\mathbf{r}_0,A}^\dagger f_{\mathbf{r}',B} + f_{\mathbf{r}_0,B}^\dagger f_{\mathbf{r}',A} \right) + h.c., \quad (A5)$$

where $\mathbf{r}'$ includes the nearest neighboring sites of $\mathbf{r}_0$. After a Schrieffer-Wolf transformation, $H_{imp} + H_{hyb}$ can be further written in the Kondo regime simply as the exchange coupling to an effective local spin-half impurity $\mathbf{S}_{imp}$ formed by the local CS fermions at $\mathbf{r}_0$:

$$H_{imp} + H_{hyb} = \sum_{\mathbf{r}} \lambda(\mathbf{r}) f_{\mathbf{r},\alpha}^\dagger \boldsymbol{\tau}_{\alpha\beta} f_{\mathbf{r},\beta} \cdot \mathbf{S}_{imp}, \quad (A6)$$

where $\boldsymbol{\tau}$ is the Pauli matrix defined in the pseudospin (sublattice) space. To be more general, we allow $\mathbf{r}$-dependence of the exchange coupling $\lambda(\mathbf{r})$, describing the scattering potential to the effective quantum impurity. We have shown that the effect of the local defect on the frustrated magnet can be cast into an effective Kondo-exchange in the pseudospin space. This leads to Eq.(8) of the main text.

## Appendix B: mapping to the impurity model coupled to 1D soft fermions

In the main text, we have shown the general procedure how to map from the impurity model in 2D Dirac fermions to that in 1D soft modes. In this section, we illustrate the detailed transformations using the honeycomb lattice model as the example.

After projection to long-wave regime near the Dirac CS valleys $\mathbf{K}$ and $\mathbf{K}'$, the bath is described as

$$H_0 = v_F \sum_{\mathbf{k}} f_{\mathbf{k},\alpha}^{(+)\dagger} (\boldsymbol{\tau}_{\alpha\beta} \cdot \mathbf{k} - \mu) f_{\mathbf{k},\beta}^{(+)} + v_F \sum_{\mathbf{k}} f_{\mathbf{k},\alpha}^{(-)\dagger} (-\boldsymbol{\tau}_{\alpha\beta}^{\mathrm{T}} \cdot \mathbf{k} - \mu) f_{\mathbf{k},\beta}^{(-)}, \quad (B1)$$

with valley-dependent PSML. $H'$ is projected into the long-wave regime as,

$$H_p' = g_d \sum_{\mathbf{k},\mathbf{k}',a=\pm} f_{\mathbf{k},\alpha}^{(a)\dagger} \boldsymbol{\tau}_{\alpha\beta} \cdot \mathbf{S}_{imp}, f_{\mathbf{k}',\beta}^{(a)} + g_t \sum_{\mathbf{k},\mathbf{k}',a=\pm} f_{\mathbf{k},\alpha}^{(a)\dagger} \boldsymbol{\tau}_{\alpha\beta} \cdot \mathbf{S}_{imp}, f_{\mathbf{k}',\beta}^{(\overline{a})}, \quad (B2)$$

where $a$ denotes the two valleys. One firstly make a unitary transformation to diagonalize CS fermions at each Dirac valley, $H_p'$ transform accordingly under the unitary transformation, leading to

$$H_p' = g_d \sum_{\mathbf{k},\mathbf{k}',a=\pm} c_{\mathbf{k}}^{(a)\dagger} U^{(a)}(\theta_{\mathbf{k}}) \tau^i U^{(a)\dagger}(\theta_{\mathbf{k}'}) c_{\mathbf{k}'}^{(a)} S_{imp}^i + g_t \sum_{\mathbf{k},\mathbf{k}',a=\pm} c_{\mathbf{k}}^{(a)\dagger} U^{(a)}(\theta_{\mathbf{k}}) \tau^i U^{(\overline{a})\dagger}(\theta_{\mathbf{k}'}) c_{\mathbf{k}'}^{(\overline{a})} S_{imp}^i, \quad (B3)$$

where $c_{\mathbf{k}}^{(a)}$ is the transformed spinor in band (sublattice) space at valley $a$, $U^{(a)}(\theta_{\mathbf{k}})$ the unitary rotation matrix applied for fermions at valley $a$ which is only dependent

on the angle of momentum $\theta_{\mathbf{k}}$. Then, utilizing the rotational symmetry of the impurity scattering, we transform the fermions to the orbital angular momentum partial waves using $c_{\mathbf{k}}^{(a)} = \sum_l e^{il\theta} c_{l,k}^{(a)}/\sqrt{2\pi k}$, where $l$ is the partial wave index. After insertion of the specific form of the unitary rotation matrix $U^{(a)}(\theta_{\mathbf{k}})$, the integral over the polar angle automatically picks up several different partial waves $l$, generating the following coupling as,

$$H_p' = g_d \sum_a \int dk dk' \sqrt{kk'} c_{l,k}^{(a)\dagger} U^{(a)}(l) \boldsymbol{\tau} \cdot \mathbf{S}_{imp} U_l^{(a)\dagger} c_{l,k'}^{(a)}$$
$$+ g_t \sum_a \int dk dk' \sqrt{kk'} c_{l,k}^{(a)\dagger} U^{(a)}(l) \boldsymbol{\tau} \cdot \mathbf{S}_{imp} U_l^{(\overline{a})\dagger} c_{l,k'}^{(\overline{a})},$$
$$(B4)$$

where constants have been absorbed into the tuning parameter $g_d$ and $g_t$, the sum over repeated notations such as $l$ is implicit. $U_l^{(a)}$ is the rotation matrix again transformed to the angular orbital momentum space, whose components are delta functions that select the channel $l$ relevant to the impurity, i.e.,

$$U_l^{(a)} = \frac{1}{\sqrt{2}} \begin{pmatrix} \delta_{l,0} & a\delta_{l,-a} \\ \delta_{l,0} & -a\delta_{l,-a} \end{pmatrix},$$
$$(B5)$$

where $a = \pm$ denotes the two valleys. Eq.(B4) implies that the impurity is coupled to an effective CS fermions $d_{\mathbf{k}}^{(a)} = \sum_l U^{(a)\dagger}(\theta_{\mathbf{k}}) c_{l,k}^{(a)}$, which is combinations of 1D CS fermions with different index $l$ for different valleys. $l = 0, -1$ are coupled to the impurity at $\mathbf{K}$ whle $l = 0, 1$ are involved at $\mathbf{K}'$ valley. Therefore, the impurity only picks up these relevant $l$ channels. Since the bath, after rotation to the angular orbital momentum space, enjoy independent $l$ components with $l$ being good quantum number due to the rotational invariance of the problem, we can select from the bath these relevant channels, leading to,

$$H_0 = \sum_{l=-1,0} \int_0^\infty dk (\epsilon_{k,\alpha} - \mu) c_{k,l,\alpha}^{(+)\dagger} c_{k,l,\alpha}^{(+)}$$
$$+ \sum_{l=0,1} \int_0^\infty dk (\epsilon_{k,\alpha} - \mu) c_{k,l,\alpha}^{(-)\dagger} c_{k,l,\alpha}^{(-)},$$
$$(B6)$$

where $\epsilon_{k,\alpha} = \alpha v_F k$. It is convenient to introduce the energy representation for the impurity problem [82], and define the effective CS fermions with combination of operators for the conduction and valence Dirac band as,

$$d_\epsilon^{(+)} =$$
$$\frac{1}{\sqrt{2}}[c_{\epsilon,0,+}^{(+)}\theta(\epsilon) + c_{\epsilon,0,-}^{(+)}\theta(-\epsilon), c_{\epsilon,-1,+}^{(+)}\theta(\epsilon) - c_{\epsilon,-1,-}^{(+)}\theta(-\epsilon)]^{\mathrm{T}},$$
$$d_\epsilon^{(-)} =$$
$$\frac{1}{\sqrt{2}}[c_{\epsilon,0,+}^{(-)}\theta(\epsilon) + c_{\epsilon,0,-}^{(-)}\theta(-\epsilon), -c_{\epsilon,1,+}^{(-)}\theta(\epsilon) + c_{\epsilon,1,-}^{(-)}\theta(-\epsilon)]^{\mathrm{T}}.$$
$$(B7)$$

Using $d_\epsilon^{(a)}$, $H_0$ is cast into a simple form as,

$$H_0 = \sum_{a,\sigma} \int_{-\infty}^\infty d\epsilon(\epsilon - \mu) d_{\epsilon,\sigma}^{(a)\dagger} d_{\epsilon,\sigma}^{(a)},$$
$$(B8)$$

where $v_F$ is set to 1. $d_{\epsilon,\sigma=1,2}^{(a)}$ are the two entries of the spinor defined in Eq.(19) and (20). Accordingly, the hybridization term $H_p'$ is reduced to the following form as,

$$H_p' = g_d \sum_a \int_{-\infty}^{+\infty} d\epsilon d\epsilon' [\rho(\epsilon)\rho(\epsilon')]^{1/2} d_\epsilon^{(a)\dagger} \boldsymbol{\tau} \cdot \mathbf{S}_{imp} d_{\epsilon'}^{(a)}$$
$$+ g_t \sum_a \int_{-\infty}^{+\infty} d\epsilon d\epsilon' [\rho(\epsilon)\rho(\epsilon')]^{1/2} d_\epsilon^{(a)\dagger} \boldsymbol{\tau} \cdot \mathbf{S}_{imp} d_{\epsilon'}^{(\overline{a})},$$
$$(B9)$$

where $\rho(\epsilon) = |\epsilon|/2\pi v_F^2$ is the density of states of Dirac CS fermions, leading to a pseudogap in the above hybridizations. Detailed studies on the pseudogapped cases have shown that the strong coupling fixed points at zero temperature are not modified by approximating the density of states by that of the Fermi energy [89], as long as $\mu \neq 0$. With this approximation, one can absorb the density of states into the couplings and rename the fermionic field as $\psi^a$. This leads to an Kondo-exchange model coupled to 1D chiral soft modes, in consistent with the general form, i.e., Eq.(3),(4) in the main text.

## Appendix C: derivation of the decoupled Wess-Zumino-Witten CFT using non-Abelian gauge invariance

The infrared fixed point of the reduced model (impurity coupled to the 1D soft modes) is described by a Wess-Zumino-Witten (WZW) CFT. We now show in this section that the underlying CFT has a decoupled multichannel structure and can be derived simply from the principle of gauge invariance. The following contents are separated into three steps including the derivation of Ward identities from the non-Abelian gauge symmetry, the chiral symmetry, and the deduction of the exact functional free energy.

### 1. Ward Identity from non-abelian gauge transformations

From the mapped 1D model of soft modes, we can start from a Dirac field in a representation $r$ of a Lie group $G$ coupled with a given gauge field $A$, namely $\mathcal{L} = \bar{\psi}(i\not{D} - m)\psi$ with $D_\mu = \partial_\mu - igA_\mu$. We define the free energy $W$ as

$$e^{-iW[A]} = Z[A] = \int \mathcal{D}\psi \mathcal{D}\bar{\psi} \, e^{iS}.$$
$$(C1)$$

The classical theory is invariant under the gauge transformations, $\psi \to U\psi, \bar{\psi} \to \bar{\psi}U^{-1}$, and $A_\mu \to A^U =$

$UA_\mu U^{-1} + \frac{i}{g} U \partial_\mu U^{-1}$, whose infinitesimal version is $\psi \to (1 + i\alpha)\psi$, $\bar{\psi} \to \bar{\psi}(1 - i\alpha)$, and $A_\mu \to A_\mu + \frac{1}{g}\mathcal{D}_\mu \alpha$. Assuming that the functional measurement $\mathcal{D}\bar{\psi}\mathcal{D}\psi$ is also gauge invariant, the free energy satisfies

$$W[A] = W[A^U] + 2\pi n[U] \qquad (C2)$$

with $n$ being an integer determined by $U$, which vanishes for infinitesimal transformations. Accordingly,

$$0 = W[A_\mu + \mathcal{D}_\mu \alpha] - W[A_\mu] = \int dx \frac{\delta W}{\delta A_\mu^a}(\mathcal{D}_\mu \alpha)^a$$

$$= -\int dx \, \mathrm{tr}\alpha \mathcal{D}_\mu \frac{\delta W}{\delta A_\mu},$$

which implies

$$\mathcal{D}_\mu \frac{\delta W}{\delta A_\mu} = 0. \qquad (C3)$$

The variation of the free energy to the gauge field is then calculated as

$$-i\frac{\delta W}{\delta A_\mu} = \frac{1}{Z}\frac{\delta Z}{\delta A_\mu} = \frac{1}{Z}\int \mathcal{D}\psi \mathcal{D}\bar{\psi} \, i\frac{\delta S}{\delta A_\mu}e^{iS}$$

$$= i\frac{1}{Z}\int \mathcal{D}\psi \mathcal{D}\bar{\psi} \, J^\mu e^{iS} = i\langle J^\mu \rangle_A.$$

Thus we prove the Ward identity

$$\mathcal{D}_\mu \langle J^\mu \rangle = 0. \qquad (C4)$$

### 2. Ward Identity from chiral invariance

Now we study the chiral gauge transformations given by $\psi \to \psi' = (1 + i\alpha\gamma^5)\psi$m $\bar{\psi} \to \bar{\psi}' = \bar{\psi}(1 + i\alpha\gamma^5)$ and $A_\mu \to A'_\mu = A_\mu + \mathcal{D}_\mu \alpha\gamma^5$. It is straightforward to check that the classical theory is invariant under the gauge transformations. However the functional measurement does not respect the transformations, leading to a Jacobian determinant $\mathcal{J}$. Thus the Ward identity should be modified because of $W[A'] \neq W[A]$. From the partition function, we obtain,

$$Z[A] = \int \mathcal{D}\psi \mathcal{D}\bar{\psi} \, e^{iS[\psi,\bar{\psi},A]} = \int \mathcal{D}\psi' \mathcal{D}\bar{\psi}' \, \mathcal{J} \, e^{iS[\psi',\bar{\psi}',A']}$$

$$= Z[A'] + \int \mathcal{D}\psi \mathcal{D}\bar{\psi} \int dx \frac{\delta \mathcal{J}}{\delta \alpha^a}\Big|_{\alpha(x)=0}\alpha^a(x)e^{iS[\psi,\bar{\psi},A']},$$

leading to

$$\frac{Z[A'] - Z[A]}{Z[A]} = -\langle \int dx \frac{\delta \mathcal{J}}{\delta \alpha^a}\Big|_{\alpha(x)=0}\alpha^a(x)\rangle. \quad (C5)$$

Besides, we have

$$\frac{Z[A'] - Z[A]}{Z[A]} = -ig\langle \int dx(\mathcal{D}_\mu J^{5\mu})^a \alpha^a\rangle. \qquad (C6)$$

Thus, conservation equation for the axial current is obtained as,

$$\mathcal{D}_\mu J^{5\mu} = -i\frac{1}{g}\frac{\delta \mathcal{J}}{\delta \alpha}\Big|_{\alpha=0}. \qquad (C7)$$

The remaining task is then to evaluate the Jacobian determinant. This can be readily done using the method developed by Fujikawa, which is also utilized in a similar situation of 3+1D with the chiral anomaly. A straightforward calculation in 1+1D then generates the Ward identity for the axial current as,

$$\mathcal{D}_\mu J^{5\mu} = -\frac{C(r)}{2\pi}\epsilon^{\mu\nu}F_{\mu\nu}. \qquad (C8)$$

where in the derivation we have defined the Dirac matrices $\gamma^0 = \sigma^2$, $\gamma^1 = i\sigma^1$ and $\gamma^3 = \gamma^0\gamma^1 = \sigma^3$ and used $\mathrm{tr}(t^a t^b) = C(r)\delta^{ab}$ with $C(r)$ a constant for eacg representation $r$ with $t^a$ the representation matrix.

### 3. The exact functional determinant in two dimensions

Noting that there exists a unique relation only in 1+1D dimensions, $\gamma^\mu \gamma^3 = -\epsilon^{\mu\nu}\gamma_\nu$, which enables us to rewrite the chiral current as $J^{3\mu} = -\epsilon^{\mu\nu}J_\nu$. Therefore, the two Ward identities derived above are collected into a united form of the CS fermion current as,

$$\mathcal{D}_\mu J^\mu = 0 \qquad (C9)$$

$$\epsilon^{\mu\nu}\mathcal{D}_\mu J_\nu = \frac{C(r)}{2\pi}\epsilon^{\mu\nu}F_{\mu\nu}. \qquad (C10)$$

Now the uniqueness of dimension two, compared with higher dimensions, lies in that the current $J^\mu$ is completely determined by the two Ward identities. Before solving the equations we first introduce the chiral coordinates, $x^+ = x^0 + x^1$, $x^- = x^0 - x^1$. In the chiral coordinates the metric $\eta$ and total anti-symmetric tensor $\epsilon$ are represented, respectively, as

$$\eta_{\mu\nu} = \begin{pmatrix} 0 & \frac{1}{2} \\ \frac{1}{2} & 0 \end{pmatrix}, \quad \epsilon^{\mu\nu} = \begin{pmatrix} 0 & 2 \\ -2 & 0 \end{pmatrix}. \qquad (C11)$$

Accordingly we define $J^+ = J^0 + J^1$, $J^- = J^0 - J^1$, and $A^+ = A^0 + A^1$, $A^- = A^0 - A^1$. In these notations the two identities can be cast into the following symmetric form,

$$\partial_+ J_- - i[A_+, J_-] = \frac{C(r)}{2\pi}F_{+-} \qquad (C12)$$

$$\partial_- J_+ - i[A_-, J_+] = \frac{C(r)}{2\pi}F_{-+}. \qquad (C13)$$

To obtain the explicit form of the solution, we introduce the expression for the gauge fields, $A_+ = ig^{-1}\partial_+ g$, $A_- = ih^{-1}\partial_- h$. with $g$ and $h$ being fields of group elements in

$G$. Then it is straightforward to check that

$$J_+ = \frac{C(r)}{2\pi}(ig^{-1}\partial_+ g - ih^{-1}\partial_+ h) \qquad (C14)$$

$$J_- = \frac{C(r)}{2\pi}(ih^{-1}\partial_- h - ig^{-1}\partial_- g), \qquad (C15)$$

are the solutions of the equations.

As promised we shall work out an explicit expression of the free energy $W[A]$, which is gauge invariant, using the fields $g$ and $h$. The field $t(x) \in G$ gives the gauge transformations,

$$
\begin{aligned}
A_+ = ig^{-1}\partial_+ g &\longrightarrow itg^{-1}\partial_+ gt^{-1} + it\partial_+ t^{-1} \\
&= i(gt^{-1})^{-1}\partial_+(gt^{-1}) \\
A_- = ih^{-1}\partial_- h &\longrightarrow ith^{-1}\partial_- ht^{-1} + it\partial_- t^{-1} \\
&= i(ht^{-1})^{-1}\partial_-(ht^{-1}), \qquad (C16)
\end{aligned}
$$

which are translated to $g$ and $h$ as

$$(g,h) \longrightarrow (g,h)t^{-1}. \qquad (C17)$$

The gauge invariance of $W[A]$ is now expressed as

$$W[g,h] = W[gt^{-1}, ht^{-1}], \qquad (C18)$$

for any field $t(x)$. So it is sufficient to work with the gauge $A_- = 0$, or equivalently $h$ constant.

$$
\begin{aligned}
\delta W = &-\frac{1}{\pi}\int dx \ \mathrm{tr}(g^{-1}\partial_- g \ \delta(g^{-1}\partial_+ g)) \\
= &-\frac{1}{\pi}\int dx \ \left(\mathrm{tr}(\partial_+\partial_- g^{-1}\delta g) - \mathrm{tr}(g^{-1}\partial_+ g \ g^{-1}\partial_- g \ g^{-1}\delta g)\right) \\
& \hspace{6cm} (C19)
\end{aligned}
$$

Noting that

$$
\begin{aligned}
&\delta \int dx \ \mathrm{tr}\partial_- g^{-1}\partial_+ g \\
= &-2\int dx \ \mathrm{tr}(\partial_+\partial_- g^{-1})\delta g \\
&+ \int dx \ \mathrm{tr}(g^{-1}\partial_+ g \ g^{-1}\partial_- g \ g^{-1}\delta g) \\
&+ \int dx \ \mathrm{tr}(g^{-1}\partial_- g \ g^{-1}\partial_+ g \ g^{-1}\delta g),
\end{aligned}
$$

we have

$$
\begin{aligned}
\delta W = &-\frac{1}{8\pi}\delta \int dx \ \mathrm{tr}(g^{-1}\partial^\mu g \ g^{-1}\partial_\mu g) \\
&+ \frac{1}{4\pi}\int dx \ \epsilon^{\mu\nu}\mathrm{tr}(g^{-1}\partial_\mu g \ g^{-1}\partial_\nu g \ g^{-1}\delta g). \quad (C20)
\end{aligned}
$$

Let us assume that $G = SU(N)$ and the spacetime manifold is compactified as $S^2$. Then it is well-known that the second term on the right hand of the above equation is a variation of a Wess-Zumino term. Thus the free energy

can be explicitly written as

$$
\begin{aligned}
W[g] = &-\frac{1}{8\pi}\int d^2x \ \mathrm{tr}(g^{-1}\partial^\mu g \ g^{-1}\partial_\mu g) \\
&+ \frac{1}{12\pi}\int d\tau d^2x \ \epsilon^{\mu\nu\rho}\mathrm{tr}(\tilde{g}^{-1}\partial_\mu\tilde{g} \ \tilde{g}^{-1}\partial_\nu\tilde{g} \ \tilde{g}^{-1}\partial_\rho\tilde{g}), \\
& \hspace{6cm} (C21)
\end{aligned}
$$

where $\tilde{g}(\tau, x)$ with $\tau \in [0,1]$ is a continuous extension of $g(x)$ with $\tilde{g}(0,x) = g(x)$ and $\tilde{g}(1,x)$ being constant.

Last, for the Dirac fields with both the spin and flavor as in Eq.(1) of the main text, the above derivation works but needs to be generalized with coupling to two non-Abelian gauge field $A_\mu$ and $B_{mu}$, resulting in the following Langrangian as,

$$
\begin{aligned}
\mathcal{L} &= \bar{\psi}^{ia}(i\gamma^\mu\partial_\mu\delta^{ij}\delta^{ab} + A_\mu^{ij}\delta^{ab} + \delta^{ij}B_\mu^{ab})\psi^{jb} \\
&= \bar{\psi}(i\gamma^\mu\partial_\mu 1_n \otimes 1_m + A_\mu \otimes 1_m + 1_n \otimes B_\mu)\psi,
\end{aligned}
$$

where accordingly we have $A_+ = ig_A^{-1}\partial_+ g_A$, $A_- = ih_A\partial_- h_A$, $B_+ = ig_B^{-1}\partial_+ g_B$, and $B_- = ih_B\partial_- h_B$, such that

$$A_+ \otimes 1_m + 1_n \otimes B_+ = ig_A^{-1} \otimes g_B^{-1}\partial_+(g_A \otimes g_B) \quad (C22)$$

$$A_- \otimes 1_m + 1_n \otimes B_- = ih_A^{-1} \otimes h_B^{-1}\partial_+(h_A \otimes h_B) \quad (C23)$$

For two arbitrary matrices $M$ and $N$, one has the following property $\mathrm{tr}(M \otimes N) = \mathrm{tr}(M)\mathrm{Tr}(N)$. Moreover, for $g \in SU(n)$, $\mathrm{tr}(g\partial_\mu g^{-1}) = 0$ since the Lie algebra consists of $n \times n$ traceless Hermitian matrices. With the above two identities, it is straightforward to derive that the following WZW emerges:

$$W[g_A \otimes g_B] = MW[g_A] + NW[g_B]. \qquad (C24)$$

This is the decoupled WZW CFT, from which one can read of the fusion rules in order to obtain the Kondo fixed points, which are discussed in the main text for both cases, i.e., with and without the off-diagonal entries of the scattering $g_{ab}$ (Eq.(4) of the main text).

### Appendix D: perturbative RG calculation of $\beta$-functions of exchange couplings

In order to determine the fixed points at the strong-coupling regime, we perform a perturbative RG calculation of the $\beta$-functions with respect to the derived effective 1D model, which reads as $H = H_0 + H'_p$, where $H_0$ is the rotated Hamiltonian in the valley space with respect to Eq.(B6), which is of the form,

$$H_0 = \sum_{m=1,2}\sum_\sigma \int_{-\infty}^{\infty} d\epsilon(\epsilon - \mu)d_{\epsilon,m,\sigma}^\dagger d_{\epsilon,m,\sigma}, \qquad (D1)$$

and $H'_p$ is approximated by using the density of states at the Fermi energy, $\rho_0$, leading to

$$H'_p = g_1 \int_{-\infty}^{\infty} d\epsilon d\epsilon' d^\dagger_{\epsilon,1,\sigma} \boldsymbol{\tau}_{\sigma\sigma'} \cdot \mathbf{S}_{imp} d_{\epsilon',1,\sigma'}$$
$$+ g_2 \int_{-\infty}^{\infty} d\epsilon d\epsilon' d^\dagger_{\epsilon,2,\sigma} \boldsymbol{\tau}_{\sigma\sigma'} \cdot \mathbf{S}_{imp} d_{\epsilon',2,\sigma'}, \quad \text{(D2)}$$

where a rotation in the valley space is performed, leading to the channel $m = 1, 2$, and $g_1 = \rho_0(g_d + g_t)$, $g_2 = \rho_0(g_d - g_t)$. The perturbative expansion over the two terms in $H'_p$ can be constructed with Feynman diagrams to the two-loop order. Integrating out the fast mode momentum leads to the renormalization group flow as,

$$dg_1/dl = g_1^2 - g_1(g_1^2 + g_2^2)g_1/2, \quad \text{(D3)}$$
$$dg_2/dl = g_2^2 - g_2(g_1^2 + g_2^2)g_1/2. \quad \text{(D4)}$$

where $dl = d\Lambda/\Lambda$ is the RG scaling parameter. The first term obtained from second order is relevant, showing the asymptotic free of the exchange coupling, and the second term from the third order contributes a suppression of the relevant flow, generating a channel-mixed fixed point with finite values of $g$'s, as shown by Fig.3(a) of the main text.

### Appendix E: The thermal conductivity from CS fermions

We now list in this section details for calculation of the thermal conductivity, with respect to both of the two Kondo-generated fixed points. The starting point is Eq.(D1) and (D2), which have been bosonized in the non-Abelian fashion before. Following [70], we introduce the left and right movers denoted by the operators $d_{L/R,m,\sigma}$, and Eq.(D1) can be transformed into an equivalent form describing a semi-infinite 1D chain. The correlation function $\langle d^\dagger_{L,m,\sigma}(z_1) d_{R,m,\sigma}(z_2) \rangle$, with $z$ being the complex coordinate, can be readily calculated. Due to the semi-infinite geometry of the 1D chain, a boundary occurs in the complex plane. For a "free" boundary (the weak coupling fixed point), the correlation function is obtained

as $\langle d^\dagger_{L,m,\sigma}(z_1) d_{R,m,\sigma}(z_2) \rangle_{Free} = 1/(z_1 - \overline{z}_2)$. Following [69–75], when the boundary is nontrivial (the Kondo fixed point), the correlation function is evaluated to be $\langle d^\dagger_L(z_1) d_R(z_2) \rangle_{Kondo} = S \langle d^\dagger_{L,m,\sigma}(z_1) d_{R,m,\sigma}(z_2) \rangle_{Free}$, where $S$ is a scattering matrix which is reduced to $S = -1$ (0) for the FL (NFL) fixed point. Furthermore, we also assume that the impurities are dilute in the studied 2D system with the impurity density $n_{imp}$. In this case, the scattering time $\tau_s$ is related to the retarded self-energy $\Sigma^R(\omega)$ via $\tau_s^{-1} = -2\text{Im}\Sigma^R(\omega)$. Note that $\Sigma^R(\omega)$ can be readily obtained from the scattering matrix $S$ [74].

The thermal current is caused by the energy transport carried by the $d$-fields, which is proportional to the particle density and the group velocity, i.e.,

$$\mathbf{j}_E = -v_F \hat{k} \sum_{m,\sigma} \int d\epsilon(\epsilon - \mu) d^\dagger_{\epsilon,m\sigma} d_{\epsilon,m,\sigma}, \quad \text{(E1)}$$

where $\hat{k} = \mathbf{k}/k$ is the unit vector along direction of $\mathbf{k}$ and $v_F$ is set to 1 in the following calculations. The thermal conductivity is related to the retarded current-current correlation function $\pi^R(\omega)$, and can be computed via $T\sigma_E(\omega) = -\text{Im}\pi^R(\omega)/\omega|_{\omega \to 0}$.

To calculate $\pi^R(\omega)$, we firstly calculate the imaginary-time current-current correlation function, $\pi(\tau) = \langle \hat{T}_\tau \mathbf{j}_E(\tau) \mathbf{j}_E(0) \rangle$, where $\hat{T}_\tau$ is the imaginary time ordering operator. Then, one firstly use Eq.(E1) and then make Fourier transformation and analytic continuation to obtain the retarded correlation function, $\pi^R(\omega)$. Finally, the thermal conductivity is obtained as

$$T\sigma_E = \frac{1}{3} \int d\epsilon \xi^2 \int d\nu \delta(\epsilon - \nu) \tau_s (-\frac{\partial}{\partial \nu} n_F(\nu)), \quad \text{(E2)}$$

where $n_F(\nu) = 1/\exp[\beta(\nu - \mu) + 1]$ denotes the Fermi distribution function. With keeping the lowest order dependence on $T$, it can be derived from Eq.(E2) that $\sigma_E(T)/T = (\pi^3 \rho_0)/[9(1 - S)n_{imp}]$. At finite temperatures, the higher-order dependence on T needs to be taken into account [74, 75]. With considering these corrections, we obtain the temperature dependence of the thermal conductivity for both the FL and the NFL fixed point, as shown by the main text.

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
