# Peer review of "Kondo signatures in defected Dirac spin liquids: Non-Abelian bosonization after Chern-Simons fermionization"

_SciPost Physics_

## Round 1 · Referee Report · Anonymous (Referee 1) · 2024-5-6

Strengths

In this paper, the authors study a particular type of defect in a Dirac quantum spin liquid by mapping the problem to the well-studied Kondo problem.

Weaknesses

  1. The setting of the problem (the particular type of defect) is not very well motivated. Perhaps the authors wanted to find a problem which can be mapped to the Kondo problem, but how it can be realized/engineered in experiments?
  2. In the authors's scheme, first the quantum spin system is handled by Chern-Simons ferminonization. The authors claim that "the gauge fluctuations are apparently suppressed" in the abstract, but I don't find any discussion in the text. Maybe the authors' position is that it was shown in earlier papers, but they should discuss it explicitly (the readers may be referred to the literature for details but still the main points should be shown) as it is rather crucial for this paper.

Report

This paper would be of a certain interest in the community as the authors develop a theoretical framework to study a defect in a quantum spin liquid. I hope that the authors will revise the paper to amend the weaknesses.

Requested changes

  1. Discuss more motivations for studying the particular type of defect, including potential experimental realizations. If there is none, would this be a good setting for numerical identification of the potential quantum spin liquid?
  2. Discuss/justify the suppression of the gauge fluctuations, which is crucial for the analysis presented in the paper.

Recommendation

Ask for major revision

---

## Round 1 · Referee Report · Anonymous (Referee 2) · 2024-6-27

Strengths

  1. The manuscript studies a problem that is of theoretical interest and is relevant to real materials.

Weaknesses

  1. The manuscript lacks theoretical clarity.
  2. Some of the discussion is technically flawed (see report).

Report

The manuscript studied the problem of Kondo impurity in the Dirac spin liquid using the Chern-Simons flux attachment and 1+1D CFT technique. They discussed several experimental consequences of their results. The physical problem they studied is interesting and is highly relevant to real materials. Nevertheless, I think some of the discussions are problematic, so I cannot recommend the manuscript for the publication in any journal before a major revision is made.

  1. The authors are using the Chern-Simons fermionization, i.e., lattice version of flux attachment in quantum Hall physics, to study Dirac spin liquid. However, this approach is problematic because the Chern-Simons fermionization breaks time-reversal and parity symmetry, such that the Dirac singlet mass (which is relevant) is not forbidden by any symmetry. As a consequence, one needs to fine tune the Dirac (singlet) mass to hit the gapless point. The theory the author studied corresponds to a phase transition (e.g. between Chiral spin liquid and Mott insulator or superfluid), not the Dirac spin liquid phase. The authors should be aware of this important issue, and make it clear in the manuscript. Also, the authors should write explicitly about the field theory they studying, namely $N_f=2$ two-component Dirac fermions couple to $U(1)_1$ Chern-Simons term.

  2. The authors state at various places (including the abstract) that 'gauge fluctuation is suppressed by the Chern-Simons term' without further justification. This statement is not correct. The gauge fluctuation would only be suppressed if the Chern-Simons level k were high, while the authors are studying k=1, for which the gauge fluctuation is still very strong. Indeed, there is no fundamental difference from the case with no Chern-Simons term, e.g., the famous Dirac spin liquid.

Since the gauge fluctuation is still strong, the authors should not treat the Kondo impurity as if it were coupled to free Dirac fermions. Without considering gauge fluctuation, the manuscript appears to present known results of Kondo impurity in free Dirac fermion theory, with only a slightly different physical interpretation.

Requested changes

  1. Clarify the manuscript is studying a quantum phase transition instead of Dirac spin liquid. Write explicitly the quantum field theory of the theory. Revise the discussions of physical relevance of their paper to the Dirac spin liquid, e.g. in the introduction. Alternatively, the author may consider to replace the Chern-Simons fermionization with other approach that is suitable for the Dirac spin liquid.

  2. Rephrase the incorrect statement about the gauge fluctuation is suppressed by Chern-Simons term. Also in the analysis they should consider to include gauge fluctuations.

Recommendation

Ask for major revision

---

## Editorial Decision

awaiting_resubmission